energy/ocean engineering/computer modelling and simulation

tidal power, renewable energy, resource assessment, uncertainty, bottom friction, bed roughness

**Author for correspondence:**
M. J. Kreitmair
e-mail: mk2040@cam.ac.uk

# The effect of bed roughness uncertainty on tidal stream power estimates for the Pentland Firth

M. J. Kreitmair[1], T. A. A. Adcock[2], A. G. L. Borthwick[1], S. Draper[3] and T. S. van den Bremer[2]

[1]School of Engineering, University of Edinburgh, Edinburgh EH9 3FB, UK
[2]Department of Engineering Science, University of Oxford, Oxford OX1 3PJ, UK
[3]Faculty of Engineering, Computing and Mathematics, The University of Western Australia, Crawley Western Australia 6009, Australia

MJK, 0000-0001-8476-0063; TAAA, 0000-0001-7556-1193;
AGLB, 0000-0001-6053-7764; SD, 0000-0002-4185-0111;
TSvdB, 0000-0001-6154-3357

Uncertainty affects estimates of the power potential of tidal currents, resulting in large ranges in values reported for sites such as the Pentland Firth, UK. Kreitmair *et al.* (2019, *R. Soc. open sci.* **6**, 180941. (doi:10.1098/rsos.191127)) have examined the effect of uncertainty in bottom friction on tidal power estimates by considering idealized theoretical models. The present paper considers the role of bottom friction uncertainty in a realistic numerical model of the Pentland Firth spanned by different fence configurations. We find that uncertainty in removable power estimates resulting from bed roughness uncertainty depends on the case considered, with relative uncertainty between 2% (for a fully spanned channel with small values of mean roughness and input uncertainty) and 44% (for an asymmetrically confined channel with high values of bed roughness and input uncertainty). Relative uncertainty in power estimates is generally smaller than (input) relative uncertainty in bottom friction by a factor of between 0.2 and 0.7, except for low turbine deployments and very high mean values of friction. This paper makes a start at quantifying uncertainty in tidal stream power estimates, and motivates further work for proper characterization of the resource, accounting for uncertainty inherent in resource modelling.

## 1. Introduction

In tidal stream power resource assessment, numerical models are used to predict extracted power from tidal flows. The results from

such models can be used to optimize the turbine configuration for maximum power extraction allowing for feedback between energy removal and natural flow conditions thereby taking into account environmental impacts [1]. For computational efficiency, large-scale tidal models are often based on the nonlinear shallow water equations, and depth-averaged models are employed. Three-dimensionality is important of course, particularly at the location of the turbines or other obstacles where the wake and recirculation zones can cause significant increase in the value of the bed roughness coefficient (perhaps by an order of magnitude over the background level, e.g. [2]). The consequent vertical mixing owing to secondary flows cannot be captured by depth-averaged models [3]. However, at the scale of the present study, vertical heterogeneity has an almost negligible effect, and so two-dimensional models are sufficiently capable of simulating bulk flow through the head-driven channels of the Pentland Firth and providing a sensible estimate of the associated energy removal [1]. In these models, the bed roughness coefficient, commonly applied uniformly throughout the flow domain, is generally used as a tuning parameter whose value is altered until predictions of velocity vectors and water levels are in some level of agreement with field observation; however, it is usually not possible to achieve perfect agreement. Consequently, there is uncertainty associated with the correct value of bed roughness coefficient that should be applied in depth-averaged models. Furthermore, the tuned value of roughness coefficient is usually well below the value that corresponds to a boundary layer flow over a rough surface (e.g. [4]). Uncertainty arising from the bed roughness coefficient, along with that from other sources (such as model limitations, lack of knowledge of bed conditions, etc.), can give rise to large discrepancies in the estimates of power available from potential sites. For example, predictions of the average power available from the Pentland Firth, UK, vary from 0.62 GW [5] to 9 GW [6]. In recent years, several sensitivity analyses have been carried out for different values of drag coefficient applied within the Pentland Firth. For example, in a power resource assessment of the Pentland Firth, Adcock et al. [7] examined the sensitivity of tidal stream power estimates to fixed, uniform values of bed friction coefficient $C_d$ in the range 0.0025–0.010. Adcock et al. found that no single value of $C_d$ produced results which matched the field measurements of both tidal phase and current magnitude, and settled on a value of $C_d = 0.0050$ as a compromise. In a similar study, Gillibrand et al. [8] varied $C_d$ from 0.004 to 0.02 and found $C_d = 0.008$ (again applied uniformly throughout the domain) gave best agreement, while acknowledging the significant spatial heterogeneity of the seabed within the Pentland Firth. The use of a single value for $C_d$, as is frequently used in tidal stream resource modelling, is itself a simplification which generates uncertainty in the model predictions.

Kreitmair et al. [9] examined the effect of uncertainty in bottom friction on tidal power estimates by introducing uncertainty as a (narrow) symmetric distribution about the mean bed roughness coefficient in three analytical models, namely those of Garrett & Cummins [10], Vennell [11] and Garrett & Cummins [12]. Using perturbation methods, Kreitmair et al. found that introduction of uncertainty in bed roughness coefficient produces changes to expected power in the range of −5 to 30% for a strait connecting two large oceans, depending on the relative strength of inertial to drag forces, and tidal farm geometry. It is reasonable to expect that these changes are likely to be smaller for deep channels such as the Pentland Firth (5–10%), where the hydrodynamics are less affected by bed resistance. Furthermore, Kreitmair et al. estimated a standard deviation in power owing to bottom friction uncertainty between 30 and 50% of mean power.

Focusing on the Pentland Firth, in this paper, we examine the effect of background friction uncertainty on estimates of the tidal power resource using a large-scale numerical model with realistic bathymetry. To this end, a numerical probability distribution transfer between bed roughness coefficient and power dissipated (effectively a numerical analogy to the 'derived distribution approach' [13–15] used in hydrological engineering) is applied to a series of tidal turbine fences spanning sections of the Pentland Firth. The tidal hydrodynamics are modelled using CG-ADCIRC, an open-source finite-element solver of the shallow water equations [16–18]. The power is estimated using a locally enhanced bed roughness coefficient approach, as used in models such as TELEMAC and MIKE 21. We examine the impact of bed roughness uncertainty on the mean power removed from the Pentland Firth and the standard deviation in power. We explore, in turn, how the effect of bed roughness uncertainty is altered by fence layout, changes in mean bed roughness coefficient (and thereby the hydrodynamic balance in the channel between inertial and drag forces), and changes in standard deviation of bed roughness uncertainty used as input.

This paper is structured as follows. Section 2 provides a brief description of the numerical model used to determine the power extracted from the Pentland Firth. A probability density function (pdf) transfer method is outlined by which it is straightforward to determine the power probability distribution from an input bed roughness probability distribution. Section 3 presents our estimates of expected power and its standard deviation for the Pentland Firth for different turbine fence configurations and mean and standard deviation of the bed roughness distribution. Conclusions are drawn in §4.

# 2. Methodology

## 2.1. Numerical model of the Pentland Firth

Numerical simulations for the Pentland Firth were performed using the Continuous Galerkin (CG) version of the ADvanced CIRCulation model (ADCIRC). ADCIRC is an open-source finite-element code used to solve the shallow water equations, following [18], for application on boundary fitted, unstructured triangular grids. The depth-averaged shallow water equations solved by CG-ADCIRC may be summarized as mass conservation:

$$\frac{\partial \zeta}{\partial t} + \frac{\partial (hu)}{\partial x} + \frac{\partial (hv)}{\partial y} = 0, \tag{2.1}$$

where $\zeta$ is the free surface elevation, $h = h_s + \zeta$ is the total depth, $h_s$ being the still water depth, $(u, v)$ are the depth-averaged velocity components in the $x$ and $y$-directions, and $t$ is time, and momentum conservation (in conservative form):

$$\frac{\partial (hu)}{\partial t} + \frac{\partial hu^2}{\partial x} + \frac{\partial (huv)}{\partial y} - fvh = -h\frac{\partial}{\partial x}\left[\frac{p}{\rho} + g(\zeta - \alpha\eta)\right] + \frac{\tau_{sx} - \tau_{bx}}{\rho} + M_x + D_x + B_x \tag{2.2}$$

and

$$\frac{\partial (hv)}{\partial t} + \frac{\partial (huv)}{\partial x} + \frac{\partial hv^2}{\partial y} + fuh = -h\frac{\partial}{\partial y}\left[\frac{p}{\rho} + g(\zeta - \alpha\eta)\right] + \frac{\tau_{sy} - \tau_{by}}{\rho} + M_y + D_y + B_y, \tag{2.3}$$

where $f$ is the Coriolis parameter, $p$ is the free surface pressure, $g$ is acceleration due to gravity, $\alpha$ is the Earth elasticity factor, $\eta$ is the Newtonian equilibrium tide potential, $\rho$ is the density of water, $M_{x,y}$ the depth-integrated momentum diffusion, $D_{x,y}$ the depth-integrated momentum dispersion, $B_{x,y}$ the depth-integrated baroclinic forcings, $\tau_{sx,y}$ the applied free surface stresses and $\tau_{bx,y}$ the stresses at the sea bed (see [18] for more details). The bed stresses are assumed to be quadratic in flow speed and can be expressed by means of an (uncertain) bed roughness coefficient $C_d$ via $\tau_b = (1/2)C_d u\sqrt{u^2 + v^2}$. In this study, only a single value for $C_d$ is applied throughout the domain, in keeping with the prevailing approach in the literature. In the derivation of these equations, it has been assumed that the flow is essentially horizontal (i.e. uniform through depth, which is reasonable for a turbulent boundary layer), vertical acceleration can be ignored, the pressure distribution is hydrostatic, the seabed slopes are relatively small, turbulent eddies across different scales can be modelled using a single eddy viscosity coefficient, and tides may be modelled as long waves.

Figure 1a,b shows the unstructured, triangular mesh that was fitted to the Pentland Firth. The mesh extends westward as far as the edge of the continental shelf in order to limit tidal reflections at the driving boundary [19]. The mesh was originally produced and validated by Adcock *et al.* [7], after convergence testing, with the size of the elements ranging from 300 m at the location of the turbine fences to 20 km far from the strait. This mesh sizing involved compromise between accuracy and computational performance.

The model was forced at the open boundary by prescribing time-varying water levels with an $M_2$ tide which dominates in this region [20]. This 'clamping' approach does not allow for changes to the water level owing to extraction of power by the tidal turbines and may therefore give inaccurate results owing to the incorrect calculation of mass flux at the open boundary [21]. However, [22] showed that little change to the natural currents at the boundary occurs upon introduction of power extraction (e.g. a current velocity increase of less than 3% for an unrealistically large deployment leading to a peak flow reduction of 25%), indicating that this boundary condition is adequate. The model was run for 2 days, following half a day of ramping up, before the average power over an $M_2$ tidal cycle was calculated.

### 2.1.1. Representation of turbines and fence configurations

Energy extraction by the turbines was implemented using a local increase in bed roughness coefficient, equal to $C_t$, over areas representing the turbine fences. Figure 1b shows the areas considered to be occupied by the turbines, indicated as strips labelled A–C. The increase in bed roughness is uniform over these areas, with each strip being approximately 1.5 km in width. Three fence configurations are considered in this paper: a 'fully spanned channel' configuration that deploys fences ABC, an 'asymmetrically confined' configuration that deploys fence A, and a 'laterally unconfined' scenario that deploys fence B only.

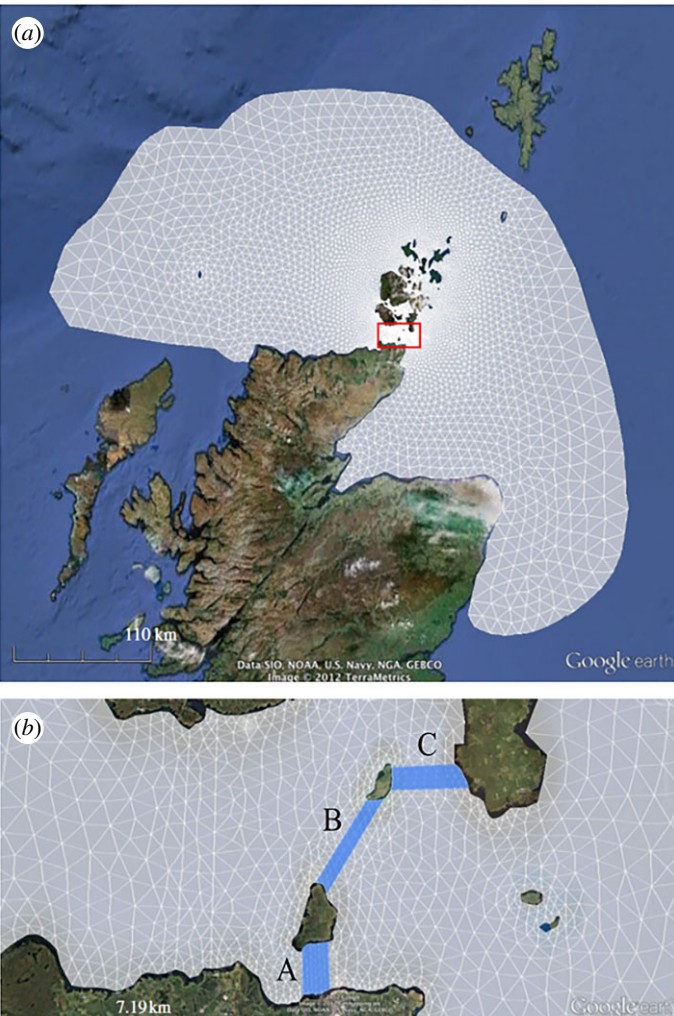

**Figure 1.** (*a*) Mesh used in numerical solver and (*b*) close-up of mesh and fences deployed at location of interest in the Pentland Firth, overlaid onto a Google Earth image of site (adapted from [7]).

The average power removed from the flow by the turbine fences over a tidal cycle is determined by calculating the dissipation due to the increase in roughness over the turbine fence area, i.e.

$$\overline{P} = \frac{1}{T}\int_0^T \left( \iint_{A_{\mathrm{turb}}} \rho C_t |\mathbf{u}|^3 \, \mathrm{d}A \right) \mathrm{d}t,$$ (2.4)

where $\rho$ is water density (taken to be 1027 kg m$^{-3}$), $T$ is the tidal period, $A_{\mathrm{turb}}$ is the plan area of enhanced bed roughness and $\mathbf{u} = (u, v)^T$ is the depth-averaged velocity vector in the presence of the roughness increase. The over-bar notation for time-averaged power is omitted from here on.

## 2.2. Uncertainty propagation method

Uncertainty in the bed roughness coefficient $C_d$ is introduced by means of a normally distributed pdf with prescribed mean and variance $\{\mu_{C_d}, \sigma^2_{C_d}\}$. A normal distribution is chosen as it is assumed that the bed roughness coefficient may be modelled as a non-skewed random variable where the causal processes underlying the coefficient are additive, i.e. bed roughness is a sum of contributions from skin friction, form drag, momentum transfer, and vegetation. The distribution is then symmetrically truncated at $C_d = 0$ (to eliminate the possibility of encountering negative $C_d$ values) and $C_d = 2\mu_{C_d}$ (to retain the same mean and symmetry properties as the parent normal distribution) and re-normalized. Consequently, the variance of the truncated pdf is reduced when compared with that of the normal distribution, and is designated by $\sigma^{*}_{C_d}{}^2$, which is termed the 'input uncertainty'.

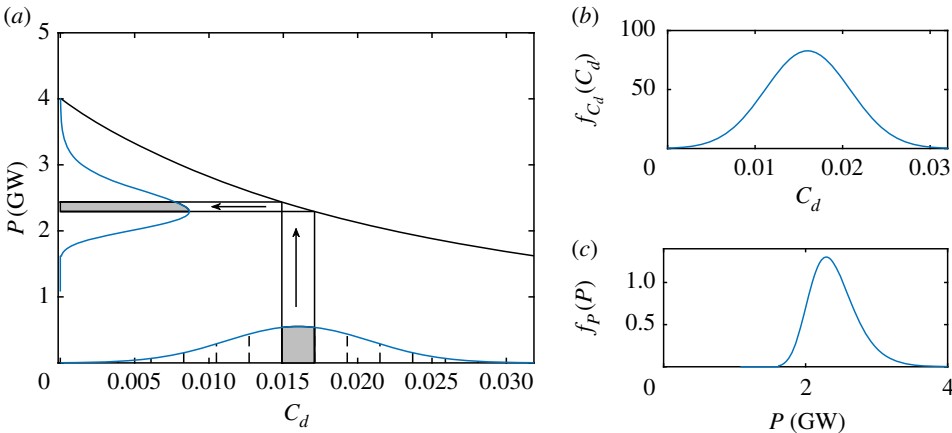

**Figure 2.** (a) Probability density transfer from a pdf for bed roughness coefficient $C_d$ to a pdf for power $P$ via a function $P = g(C_d, C_t = C_t^*)$; (b) symmetrically truncated normal distribution for $C_d$ (input); and (c) corresponding pdf for $P$ after transfer through power function (output).

To explore the impact of changes in mean bed roughness coefficient of the channel three values are considered $\mu_{C_d} = \{0.0025, 0.005, 0.016\}$, which we designate as low, medium and high channel drag scenarios, respectively. These values are chosen from the literature, where the low value is often used in depth-averaged tidal energy models (e.g. [23–25]). The high value of $\mu_{C_d} = 0.016$ was taken from [4], where it constitutes a 'best guess' for the true bed roughness coefficient of the Pentland Firth.

The pdf in bed roughness coefficient $f_{C_d}(C_d)$ is used to generate a corresponding pdf in power $f_P(P)$ by propagating $f_{C_d}$ through power as a function of bed roughness coefficient and turbine drag coefficient, $P = g(C_d, C_t)$. The procedure is as follows. For each value of turbine drag $C_t$, the pdf in $C_d$ is divided into bins of equal width and the probability of a realization of $C_d = C_{d,i}$ falling within each bin is determined by integrating over the width of the bin. This gives the associated probability of a realization of power $P = P_i = g(C_{d,i}, C_t)$, as well as the associated probability density in power at this value through $f_P(P_i) = f_{C_d}(C_{d,i}) \Delta C_{d,i} / \Delta P_i$, where $\Delta C_{d,i}$ is the width of the bin in $C_d$ and $\Delta P_i$ the associated width in $P$. Having determined the probability associated with a given value of power, the mean is given by $E[P] = \sum_i P_i(C_d = C_{d,i}) \Pr(C_d = C_{d,i})$, and the $n$th-order (centred) statistical moments may be calculated using

$$\mu_{P,n} = \sum_i (P_i(C_d = C_{d,i}) - E[P])^n \Pr(C_d = C_{d,i}), \tag{2.5}$$

where $n = 2$ gives the variance, and $n = 3$ and $n = 4$ the skewness and kurtosis, respectively.

It may be noted that the pdf for power resulting from the uncertainty transfer is no longer a normal distribution when the function $P = g(C_d)$ is nonlinear. Figure 2a shows the transfer of a symmetrically truncated normal pdf in $C_d$ (shown in figure 2b) to the corresponding pdf in power (figure 2c). Though the input pdf is symmetric, the output pdf exhibits a small positive skew owing to the nonlinear transfer.

# 3. Results

Power surfaces as a function of bed roughness coefficient $C_d$ and turbine drag coefficient $C_t$, produced from many runs of the numerical model, were used as transfer functions for the pdf. It should be emphasized that the surfaces were determined systematically over a range of $C_d$ and $C_t$ values, and cubic spline interpolation was used to refine the resolution of the surfaces. This method only required of the order of 2000 runs of the model to obtain convergence (fewer runs than Monte Carlo simulations). Examples of the surfaces and corresponding contour plots obtained for the three fence configurations are included for completeness in figure 7 in appendix A. For each value of $C_t$, the corresponding curve in $C_d$ is used as the transfer function to determine expected values for power and associated standard deviations as functions of turbine drag $C_t$. The effect of uncertainty in $C_d$ on these parameters is now explored.

## 3.1. Expected power

Figure 3 shows values of deterministic power $P$ and expected power $E[P]$ (left-hand axes) as functions of turbine drag coefficient scaled by mean bed roughness coefficient $C_t / \mu_{C_d}$ for fully spanned, asymmetrically

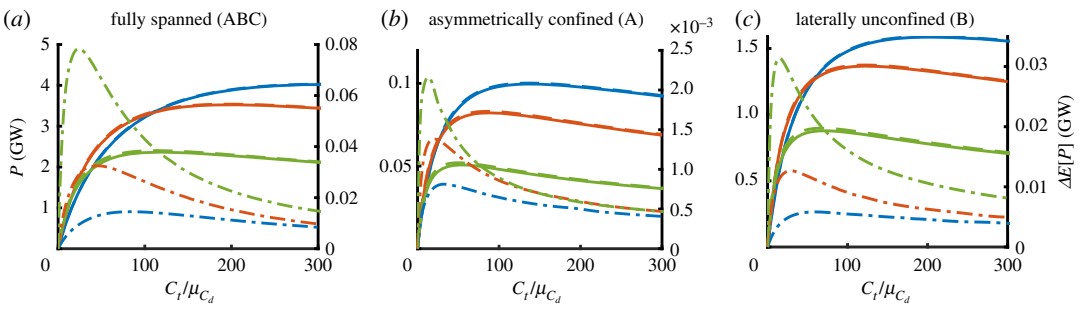

**Figure 3.** Deterministic and expected power (continuous and dashed lines, left-hand axis) and change in expected power (dashed-dotted lines, right-hand axis) in GW extracted in the fully spanned (*a*), asymmetrically confined (*b*), and laterally confined (*c*) fence deployment scenarios as a function of scaled turbine drag coefficient $C_t/\mu_{C_d}$ for different values of mean bed roughness coefficient $\mu_{C_d} = \{0.0025, 0.005, 0.016\}$ (blue, red and green, respectively). The input relative standard deviation in all cases is $\hat{\sigma}^*_{C_d} = 0.3$.

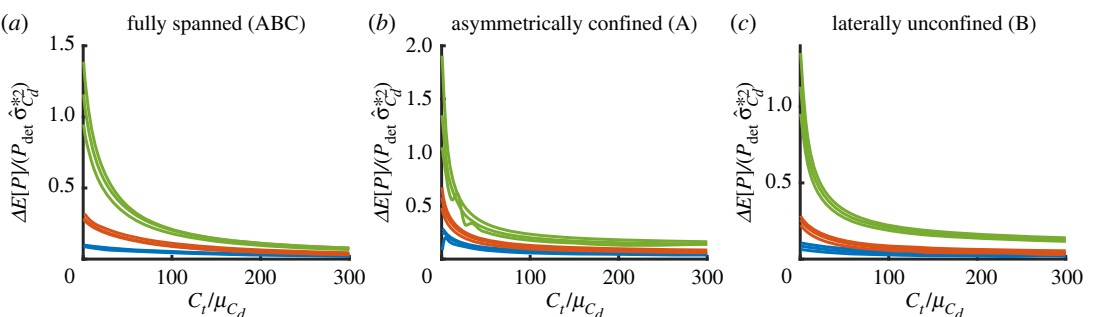

**Figure 4.** Relative change in expected power scaled by the relative input variance in the fully spanned (*a*), asymmetrically confined (*b*), and laterally confined (*c*) fence deployment scenarios as a function of scaled turbine drag for different values of mean bed roughness coefficient $\mu_{C_d} = \{0.0025, 0.005, 0.016\}$ (blue, red and green, respectively). Curves are plotted for different input relative standard deviations of $\hat{\sigma}_{C_d} = \{0.1, 0.3, 0.5\}$.

confined and laterally unconfined fence configurations. The change in expected power $\Delta E[P]$ is shown on the right-hand axes. The results shown in different colours indicate different mean bed roughness coefficients, $\mu_{C_d} = \{0.0025, 0.005, 0.016\}$, corresponding to 'low', 'medium' and 'high' channel drag scenarios. Results are shown for an input relative uncertainty of $\hat{\sigma}^*_{C_d} = 0.3$. A range of $\hat{\sigma}^*_{C_d}$ values is considered below to explore the effect of different input uncertainty values, chosen following [9]. For all values of $\mu_{C_d}$, uncertainty in $C_d$ acts to increase the expected power slightly above the deterministic value. That is, the mean power determined from a distribution of bed roughness coefficients is greater than the power determined from the mean bed roughness coefficient. This is a result of the convex dependence of power on bed roughness (i.e. Jensen's inequality). Channels with greater mean bed roughness coefficient values exhibit a larger increase in the expected power. This is not simply owing to the fact that these channels have a greater $\sigma^*_{C_d}$ for a given input relative standard deviation $\hat{\sigma}^*_{C_d} \equiv \sigma^*_{C_d}/\mu_{C_d}$, but because the relative importance of bed roughness in the dynamic balance of the channel grows with increasing bed roughness. The more $C_d$ dominates the dynamic balance (and hence the flow velocities) in a channel, the greater the impact of uncertainty on power. This may be seen more clearly from figure 4, which shows how the relative change in expected power divided by input relative variance, i.e. $\Delta E[P]/(P_{\text{det}}\hat{\sigma}^2_{C_d})$, varies as a function of turbine drag scaled by mean bed roughness coefficient. As before, different colours indicate 'low', 'medium' and 'high' channel drag scenarios. Three lines have been plotted for each $\mu_{C_d}$ corresponding to different values of input relative standard deviation of $\hat{\sigma}_{C_d} = \{0.1, 0.3, 0.5\}$. While there are some changes with increasing values of $\hat{\sigma}_{C_d}$ (higher values giving slightly higher values of $\Delta E[P]/(P_{\text{det}}\hat{\sigma}^2_{C_d})$), these changes are small. This indicates that changes in expected power owing to uncertainty are dominated by leading-order effects, despite the complexity and nonlinearity of the numerical model. In their study of the effects of uncertainty in analytic models of tidal stream power, Kreitmair *et al.* [9] found that changes in $\Delta E[P]/(P_{\text{det}}\hat{\sigma}^2_{C_d})$ are a result of higher-order statistical moments of the input pdf. The small changes in $\Delta E[P]/(P_{\text{det}}\hat{\sigma}^2_{C_d})$ indicate that these higher-order terms do not contribute greatly to variations in expected power.

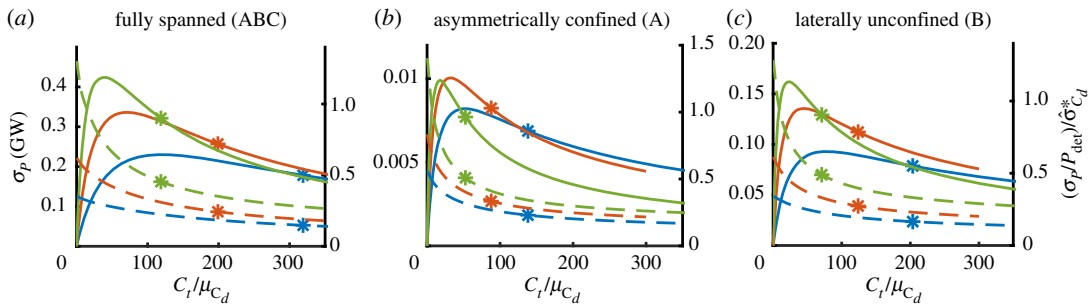

**Figure 5.** Standard deviation in power removed in GW (left-hand axes) and ratio of relative standard deviation in power to relative input standard deviation (right-hand axes) in the fully spanned (*a*), asymmetrically confined (*b*), and laterally confined (*c*) fence deployment scenarios as a function of scaled turbine drag coefficient $C_t/\mu_{C_d}$ for different values of mean bed roughness coefficient $\mu_{C_d} = \{0.0025, 0.005, 0.016\}$ (blue, red and green, respectively). The input relative standard deviation in all cases is $\hat{\sigma}^*_{C_d} = 0.3$. Asterisks indicate the location of (deterministic) optimal turbine drag.

Comparison of behaviour for different fence layouts shows that there is little qualitative difference caused by confinement of turbine fences. In all deployment scenarios, the change in expected power increases with the mean bed roughness coefficient applied throughout the channel, and does so by approximately the same proportion per unit input relative variance (figure 4). This suggests that there is little bypass flow around the deployed turbines, even in the unconfined scenario where only fence B is deployed. Of note is that the scenario with the shortest fence configuration (fence A deployed) exhibits the greatest sensitivity to bed roughness uncertainty, followed by the next shortest (fence B), and then the longest (fences ABC). This indicates that, the greater the total force due to turbine drag in the channel, (and hence the importance of bed friction on dynamic balance in the channel), the smaller the effect of uncertainty in bed roughness.

The main conclusion of this section is that changes in expected power are small, and only become significant at very low values of turbine drag, i.e. few turbines deployed. In such instances, the change in expected power can be a significant fraction of the deterministic power. For example, from figure 4*b* it can be determined that the change varies between 0.2% of deterministic power for a low channel drag scenario to 50% for a high channel drag scenario as the deployed turbine drag approaches zero. However, near the optimum the change in expected power is of the order of 3% of deterministic power for the high channel drag scenario of the asymmetrically confined fence deployment configuration (the fence configuration most sensitive to uncertainty). While the effects on extracted power are small for any but low values of turbine drag, the impact of uncertain bed roughness may be more important for available power, i.e. the fraction of power available to the turbines for power generation [7,26], because available power is optimized at total resistances lower than extractable power.

It may be noted that the results shown in figure 4 are in good qualitative agreement with fig. 4a of [9], suggesting that the analytic models of Garrett & Cummins [10] and Vennell [11] used for analysis may give good approximations to a more in-depth numerical analysis of bed roughness uncertainty effects, provided an appropriate value for the non-dimensional channel drag parameter $\lambda_0$ is determined.

## 3.2. Standard deviation in power

Figure 5 shows the standard deviation in power $\sigma_P$ (left-hand axis) and the ratio of relative standard deviation in power to that in bed roughness, i.e. $(\sigma_P/P_{\text{det}})/\hat{\sigma}^*_{C_d}$, (right-hand axis) as functions of turbine drag coefficient scaled by mean bed roughness coefficient $C_t/\mu_{C_d}$, with the colours indicating different values for mean bed roughness coefficient as before. In the cases shown, the input relative standard deviation is $\hat{\sigma}^*_{C_d} = 0.3$. Asterisks on the curves indicate the optimal value of $C_t/\mu_{C_d}$ (in the deterministic cases), i.e. the relative turbine drag which maximizes power removed by the deployed fences before introduction of uncertainty, as determined from the power surfaces in figure 7. It is evident from this figure that the maximum standard deviation in power occurs at below-optimum turbine deployments. It is further evident that different fence configurations do not affect the impact of bed roughness uncertainty differently, as may be observed from the qualitatively similar behaviour across all the configurations. The ratio of the relative uncertainties $\hat{\sigma}_P/\hat{\sigma}^*_{C_d}$ gives the factor by which the relative input uncertainty is decreased or increased upon transfer through the power function.

**Table 1.** Standard deviation in power and ratio of relative standard deviation in power to relative input standard deviation for three different values of $\hat{\sigma}^*_{C_d} = \{0.1, 0.3, 0.5\}$ at optimal and half-optimal turbine drag values (columns) and different fence configurations and mean bed roughness coefficients (rows).

| | | $\sigma_P$ (GW) | | $\sigma_P/P_{det}$ (%) | |
|---|---|---|---|---|---|
| | | $1/2C^*_{t,det}$ | $C^*_{t,det}$ | $1/2C^*_{t,det}$ | $C^*_{t,det}$ |
| ABC | $\mu_{C_d} = 0.0025$ | 0.07, 0.23, 0.4 | 0.06, 0.18, 0.3 | 2, 6, 10 | 2, 5, 8 |
| fully | $\mu_{C_d} = 0.005$ | 0.1, 0.33, 0.55 | 0.09, 0.26, 0.43 | 3, 10, 17 | 2, 7, 13 |
| spanned | $\mu_{C_d} = 0.016$ | 0.13, 0.41, 0.71 | 0.11, 0.32, 0.55 | 6, 19, 35 | 4, 14, 24 |
| A | $\mu_{C_d} = 0.0025$ | 0.003, 0.008, 0.014 | 0.002, 0.007, 0.012 | 3, 9, 16 | 2, 7, 12 |
| asymmetrically | $\mu_{C_d} = 0.005$ | 0.003, 0.01, 0.017 | 0.003, 0.008, 0.014 | 4, 14, 24 | 3, 11, 19 |
| confined | $\mu_{C_d} = 0.016$ | 0.003, 0.01, 0.017 | 0.0025, 0.008, 0.014 | 6, 22, 44 | 5, 17, 32 |
| B | $\mu_{C_d} = 0.0025$ | 0.031, 0.09, 0.15 | 0.026, 0.08, 0.13 | 2, 6, 12 | 2, 5, 9 |
| laterally | $\mu_{C_d} = 0.005$ | 0.044, 0.13, 0.22 | 0.037, 0.11, 0.18 | 3, 11, 19 | 3, 8, 15 |
| unconfined | $\mu_{C_d} = 0.016$ | 0.050, 0.16, 0.27 | 0.042, 0.13, 0.22 | 6, 20, 38 | 5, 16, 28 |

**Table 2.** Ratio of relative standard deviation in power to relative input standard deviation for three different values of $\hat{\sigma}^*_{C_d} = \{0.1, 0.3, 0.5\}$ at optimal and half-optimal turbine drag values (columns) and different fence configurations and mean bed roughness coefficients (rows).

| | | $\hat{\sigma}_P/\hat{\sigma}^*_{C_d}$ (%) | |
|---|---|---|---|
| | | $1/2C^*_{t,det}$ | $C^*_{t,det}$ |
| ABC | $\mu_{C_d} = 0.0025$ | 20, 20, 20 | 15, 15, 15 |
| fully | $\mu_{C_d} = 0.005$ | 33, 33, 34 | 24, 24, 24 |
| spanned | $\mu_{C_d} = 0.016$ | 60, 62, 64 | 44, 45, 46 |
| A | $\mu_{C_d} = 0.0025$ | 29, 29, 29 | 23, 23, 23 |
| asymmetrically | $\mu_{C_d} = 0.005$ | 42, 43, 44 | 33, 33, 34 |
| confined | $\mu_{C_d} = 0.016$ | 64, 67, 73 | 49, 51, 54 |
| B | $\mu_{C_d} = 0.0025$ | 21, 21, 20 | 17, 17, 16 |
| laterally | $\mu_{C_d} = 0.005$ | 34, 35, 34 | 27, 27, 27 |
| unconfined | $\mu_{C_d} = 0.016$ | 61, 63, 67 | 48, 49, 51 |

This ratio is smallest for small values of $\mu_{C_d}$ but increases to above unity for low values of $C_t/\mu_{C_d}$ (low turbine deployments) for the high value of the bed roughness coefficient $\mu_{C_d} = 0.016$ (dashed green lines). This is owing to the shallow gradient of the power curve with respect to bed roughness coefficient $C_d$ generally acting to decrease uncertainty upon transfer through to power. However, at low levels of turbine deployment and in channels with high mean bed roughness (i.e. long and shallow channels) uncertainty is amplified as the gradient with respect to $C_d$ is steeper. The results shown on the right-hand axis of figure 5 are in good agreement with results shown in fig. 5 of Kreitmair *et al.* [9], again dependent on choice of value for $\lambda_0$.

Quantitative values for standard deviation are given in table 1 (left columns show absolute values and right columns give standard deviation as a fraction of the deterministic power). These are given at optimal and half-optimal turbine drag values, determined for a given combination of fence configuration, mean bed roughness coefficient, and input uncertainty. For cases of high input uncertainty ($\hat{\sigma}_{C_d} = 0.5$), the standard deviation becomes a significant fraction of the deterministic power, up to 44% in the high mean channel drag scenario applied to the asymmetrically confined turbine deployment. However, for a fully spanned channel with small input uncertainty, the standard deviation in power is as low as 2% of deterministic power.

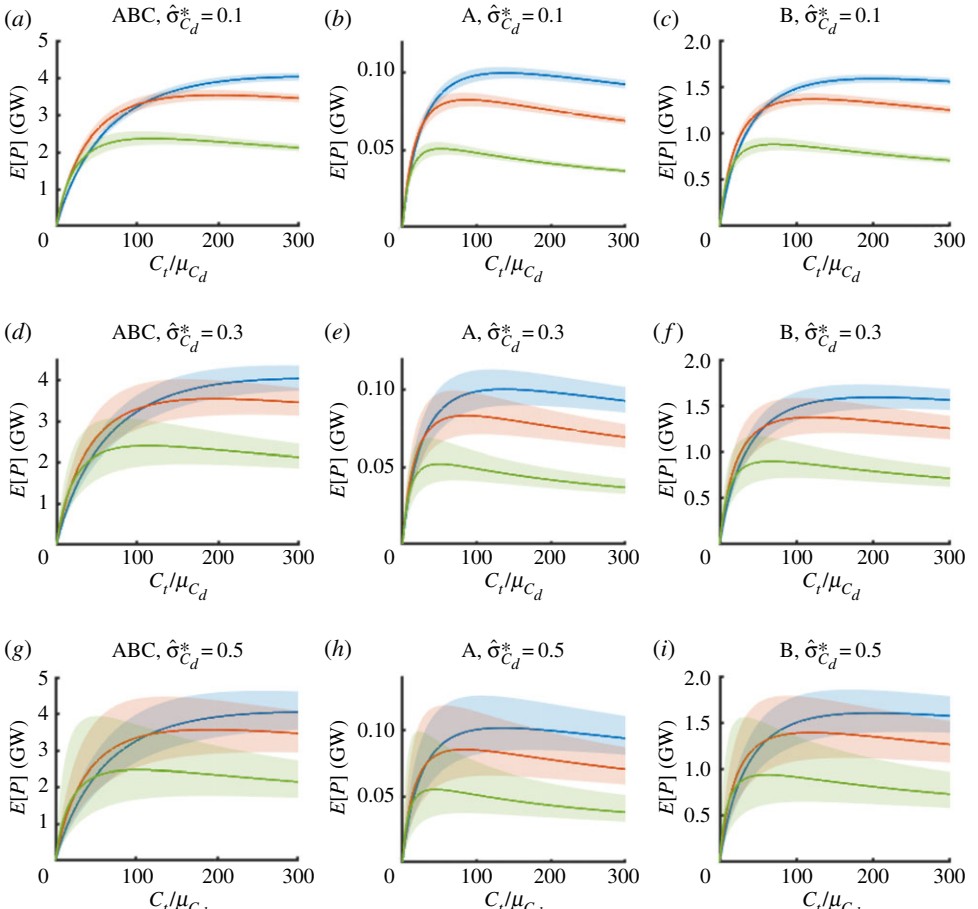

**Figure 6.** Expected power removed and 90% confidence levels (shaded) in the fully spanned (*a,d,g*), asymmetrically confined (*b,e,h*) and laterally confined (*c,f,i*) fence deployment scenarios as a function of scaled turbine drag coefficient $C_t / \mu_{C_d}$ for different values of mean bed roughness coefficient $\mu_{C_d} = \{0.0025, 0.005, 0.016\}$ (blue, red and green, respectively) and input relative standard deviation $\hat{\sigma}_{C_d} = \{0.1, 0.3, 0.5\}$ (top to bottom).

Table 2 lists the ratio of the output relative standard deviation $\hat{\sigma}_P = \sigma_P/P_{\text{det}}$ to the input relative standard deviation $\hat{\sigma}_{C_d}$. It is evident that this ratio remains largely unchanged as input uncertainty is varied, i.e. the value stays the same across the input relative standard deviation values of $\hat{\sigma}_{C_d}^* = \{0.1, 0.3, 0.5\}$. Significant (on the order of a few per cent) changes in the ratio occur only for large values of mean bed roughness coefficient, as noted before. This further confirms the result of §3.1 that the statistical behaviour is dominated by leading order effects and higher order effects begin to contribute appreciably only as $\mu_{C_d}$ is increased. Notably, however, near the optimum uncertainty is generally reduced. In practice, any realistic deployment of tidal turbines will not be near the optimum for extracted power. This is both because it would lead to unacceptable environmental change and because of the diminishing returns as more turbines are added [27].

## 3.3. Confidence intervals

The expected power from the different fence configurations is plotted in figure 6 with shaded regions illustrating the 90% confidence bands as functions of the relative turbine drag for input relative standard deviations of $\hat{\sigma}_{C_d} = \{0.1, 0.3, 0.5\}$ (rows of panels). Needless to say, with growing input uncertainty, the resulting confidence limits on power widen and begin to span a significant range. For example, in the very extreme case of high input uncertainty and high bed roughness for the fully spanned channel (green line, panel *g*) the confidence limits at optimum span as much as 2 GW for an expected power of 2.3 GW. This covers a significant portion of the range of power estimates quoted in the introduction (0.62-9 GW).

# 4. Conclusion

In this paper, we have explored the effect of bed friction uncertainty on power estimates of removable power from tidal stream turbines deployed in the Pentland Firth. Having previously examined this using perturbation methods in three idealized tidal stream power models [9], the present study analyses the effect of an uncertain bed roughness coefficient in a realistic model for the Pentland Firth. To this end, we have used a numerical method for transferring the probability distribution of an input variable through a numerically generated surface. While sensitivity studies have been conducted in the past exploring the effect of changes to bed roughness coefficient on extracted power, the approach presented in this paper is able to determine the effect of a distribution of coefficient values, giving rise to changes in the mean power, a measure of standard deviation in power values, and confidence bands. Higher-order statistical moments can easily be calculated using this method and may be of interest in other applications including flood risk.

Our conclusions are as follows. The effect of uncertainty on expected power is small except for low values of turbine drag or for high values of mean bed roughness coefficient. Near optimal levels of turbine deployment, the increase is as little as 3% even for the most sensitive fence configuration. This echoes the conclusion of the previous study by Kreitmair et al. [9] which stated that for a channel reminiscent of the Pentland Firth, the change in expected power would only be on the order of a few per cent. The previous study further concluded that about ±3 GW could be attributed to bed roughness uncertainty. The present paper limits this estimate to ±1 GW at the 90% confidence limit in the extreme case of high input relative uncertainty of $\hat{\sigma}_{C_d} = 0.5$ and high mean channel drag. Comparing this interval with the range of power estimates for the Pentland Firth from the Introduction, i.e. 0.62–9 GW, only a fraction of this range may be attributed to bed roughness uncertainty. The greater proportion is most likely owing to differences in models used, which indicates the requirement for a coherent methodology in the tidal stream energy industry to allow for comprehensive assessments to be made. It should be noted that the present paper deals with removable power rather than extractable power. However, assuming a linear relationship between the extractable and removable power (i.e. the amount of power that may be extracted is a constant fraction of the power removed from the channel), the proportional effect of bed roughness uncertainty would simply transfer to removable power. More complicated, nonlinear relationships would require further analysis.

While this study is an improvement to previous work by exploring uncertainty for a more realistic channel model, limitations remain. The model used represents turbines as a locally enhanced bed roughness coefficient smeared out over a number of grid cells, and so reproduces the aggregate effect of the turbines as they span the channel or subchannels. Hence, any energy losses in the mixing of the by-pass flow are not considered and only removable power is explored. Future work would take into account the effect of uncertainty on extractable power (i.e. power available to the turbines) rather than removable power (total power lost owing to the presence of the turbines) by incorporating the turbines in more realistic ways, such as actuator disc theory or constrained blade element momentum [28]. Furthermore, in the model used, only a single value of bed roughness coefficient is applied throughout the numerical domain. It would be of interest to explore the effect of spatially (and temporally) varying coefficient values, and the uncertainty of these, and how the value of bed roughness coefficient may be inferred from sparse measurement data.

In this paper, we have considered solely parameter uncertainty and how this varies for a given model configuration (i.e. model scale, mesh refinement, dimensionality, etc.). A further source of uncertainty is that of model discrepancy, which is the uncertainty introduced by the fact that no model is a perfect representation of a real process. Changes to the model will affect the value of calibrated bed roughness coefficient and magnitude of the associated parameter uncertainty (e.g. a finer mesh can resolve physical features which are treated by the roughness coefficient in a coarser mesh). Comparison of the two types of uncertainty requires methods which can account for both parameter and model uncertainty, such as the Kennedy-O'Hagan framework [29], some of which are already used in other fields, for instance, building energy simulation (e.g. [30]).

Data accessibility. The bathymetric data used for the numerical simulations are subject to commercial licensing restrictions. Power data obtained from numerical runs are available from the Dryad Digital Repository: https://dx.doi.org/10.5061/dryad.k3c16gt [31].

Authors' contributions. M.J.K. and T.S.v.d.B. developed the numerical method for probability transfer. The numerical model was built and verified by S.D. and T.A.A.A., with computations performed by M.J.K. Results were interpreted by M.J.K., T.S.v.d.B. and A.G.L.B. M.J.K. and T.S.v.d.B. wrote the paper. All authors gave final approval for publication.

Competing interests. We declare we have no competing interests.

Funding. M.J.K. was supported by Engineering and Physical Sciences Research Council (EPSRC) grant no. R44708. T.S.v.d.B. was supported by a Royal Academy of Engineering Research Fellowship.

Acknowledgements. All contributors have been included as authors. The authors thank Paolo Perona at the University of Edinburgh for pointing out the use of the derived distribution approach in other fields.

## Appendix A. Power surfaces

Power surfaces and contours calculated from the numerical model of the Pentland Firth are shown in figure 7 as functions of bed roughness coefficient $C_d$ and deployed turbine drag coefficient $C_t$. Each row shows the results for a given fence configuration.

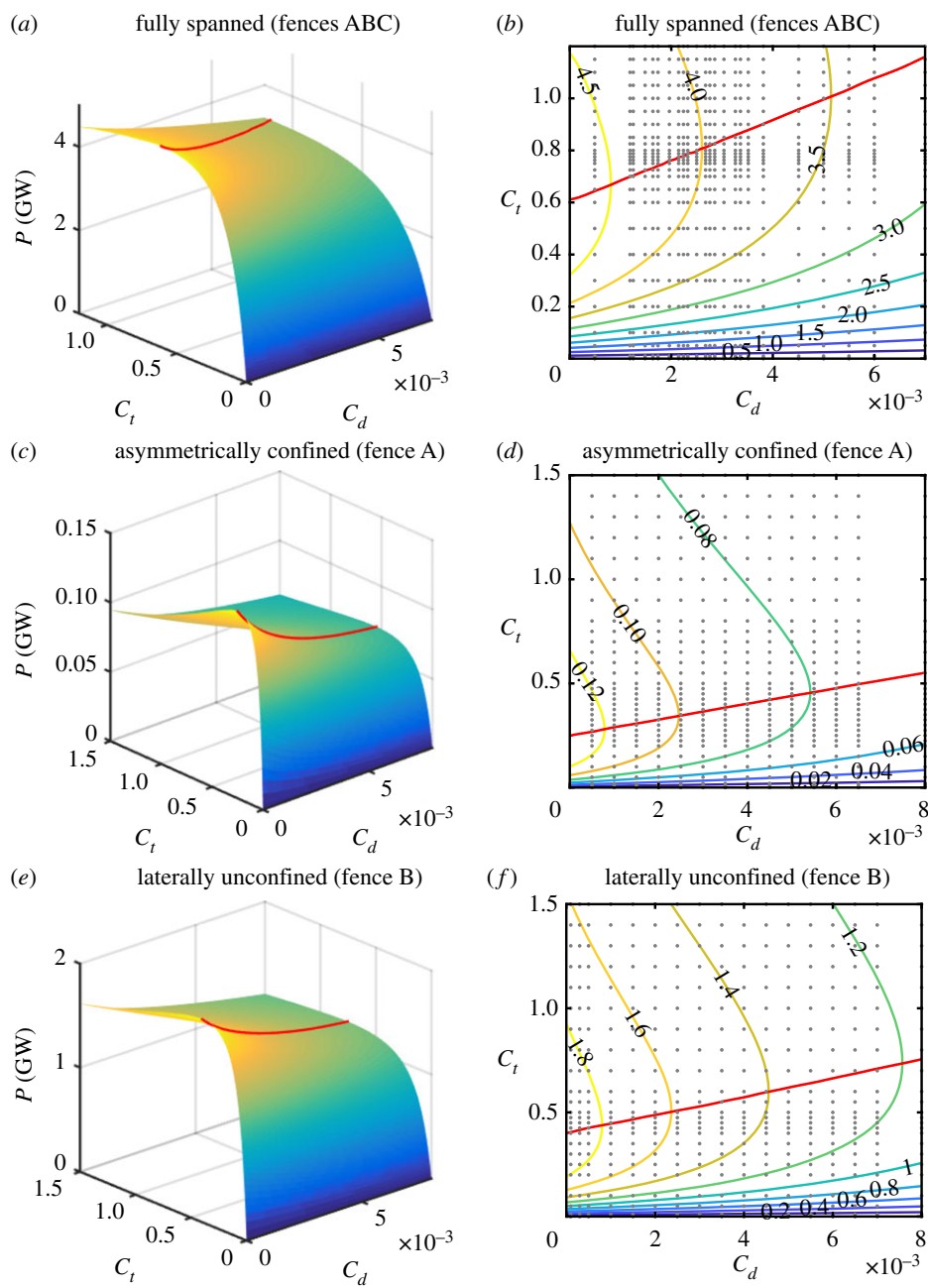

**Figure 7.** Power extracted by fences ABC (*a,b*), fence A (*c,d*) and fence B (*e,f*) as a function of bed roughness coefficient $C_d$ and turbine drag coefficient $C_t$. The red line indicates the location of maximum power dissipation. The input ($C_d$, $C_t$) values used to perform the numerical runs are indicated as grey points overlaid onto the contours.

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
