## [Reviewer comments · Royal Society Open Science]

Review History

RSOS-191127.R0 (Original submission)

Review form: Reviewer 1

Is the manuscript scientifically sound in its present form?

Yes

Are the interpretations and conclusions justified by the results?

Yes

Is the language acceptable?

Yes

Do you have any ethical concerns with this paper?

No

Have you any concerns about statistical analyses in this paper?

No

Recommendation?

Accept with minor revision (please list in comments)

Comments to the Author(s)

Manuscript ID RSOS-191127 entitled "The effect of bed roughness uncertainty on tidal stream power estimates for the Pentland Firth" for Royal Society Open Science.

The manuscript describes a study examining the impact of varying bed roughness in a real-world tidal energy setting. It uses 2D shallow water model solutions to examine the impact on energy extraction through variation in bottom friction.

It is a useful result if very incremental, so in that sense – fits the purpose of the journal.

It is clearly written.

A number of points really should be followed up upon to consolidate the worth of the manuscript.

The results appear to suggest minimal impact of varying C_d – but the same value of C_d can have a difference effect in different model configurations (scale etc). How does this sensitivity compare?

I really struggle with 2D perspectives on anything other than the very broad scale. The energy extraction will be highly heterogeneous in the vertical. How important is this likely to be?

By the same token, does the C_d variation fall within the impact of moving to 3D simulation?

Further, what's the meaning of 3% variation in power production in the context of the model. Is it thought that the present model is less than 3% from "truth"?

Is there some evidence that the present result is extendable to other systems and why?

How does the conclusion compare with the varying predictions mentioned in the abstract?

Review form: Reviewer 2

Is the manuscript scientifically sound in its present form?

Yes

Are the interpretations and conclusions justified by the results?

Yes

Is the language acceptable?

Yes

Do you have any ethical concerns with this paper?

No

Have you any concerns about statistical analyses in this paper?

No

Recommendation?

Accept with minor revision (please list in comments)

Comments to the Author(s)

This is a well written paper about tidal power in the Pentland Firth following a previous paper using idealised models. This paper uses more realistic depth-averaged shallow water tidal model for the region. I do have some comments. The paper refers to power potential but it is not clear from the start that is about 'removable power' rather than 'extractable power' clarified later. Initially I thought it was about power extracted by turbines. I realise this cannot be simply undertaken but some estimate could be given, e.g. 50% based on some idealised cases. If it was 10% for example the exercise would be rather pointless. In the Conclusions powers of GW level are discussed but this is for removable power.

Going through the paper, in the Introduction it is stated that 'there is uncertainty associated with the correct value of bed roughness coefficient that should be applied.' It should be clear that this is associated with depth-averaged models used here. 3-D models with hydrostatic pressure are more accurate and indeed show that 2-D depth-averaged models with strong curvature, recirculating flows in the extreme, can be rather misleading (Stansby et 2016). Soon after values of Cd for a rough surface are referred to but I think these are for steady flow. In oscillatory they can be different. This should be clarified.

In section 2a it is said that 'little change to the natural currents at the boundary occurs'. Little should be quantified in some way, e.g. 1% or 10% in change in tidal level or volume flux. 'little' means different things to different people!

I would like to see fig 1c enlarged so the fences can be more easily seen. Why not include in fig 1b?

In eq 2.4 is \bar{u} with or without the fence present?

The actual analysis of uncertainty is quite comprehensive and dense.

With these relatively minor points addressed I think the paper will be suitable for publication.

Stansby, P. Chini, N. and Lloyd, P. 2016 Oscillatory flows around a headland by 3D modelling with hydrostatic pressure and implicit bed shear stress comparing with experiment and depth-averaged modelling, Coastal Engineering 116, 1-14

Review form: Reviewer 3

Is the manuscript scientifically sound in its present form?

Yes

Are the interpretations and conclusions justified by the results?

Yes

Is the language acceptable?

Yes

Do you have any ethical concerns with this paper?

No

Have you any concerns about statistical analyses in this paper?

No

Recommendation?

Accept with minor revision (please list in comments)

Comments to the Author(s)

Tidal stream power estimates often rely on the results of hydrodynamic numerical models that are calibrated using bed friction coefficients. This way of doing is subjected to high uncertainties. Interestingly, this paper evaluates the effect of these uncertainties on the resource assessment by considering different scenarios of turbine deployments.

The paper, written following the IMRAD structure, is well written and contains high quality figures. The methodology is clearly described and the conclusions are well supported by the results. I therefore strongly recommend its publication.

My only (minor) comment concerns the omission of modelling studies with non-uniform roughness coefficients (in the introduction, you suggest that, in existing studies, C_d is always uniform). As using a uniform C_d is not always applicable, especially in tidal sites where the seabed is highly heterogeneous, I wonder how is the seabed in the Pentland Firth. Showing a sedimentary map could help justifying the assumption $C_d = \text{constant}$.

Decision letter (RSOS-191127.R0)

17-Oct-2019

Dear Dr Kreitmair

On behalf of the Editors, I am pleased to inform you that your Manuscript RSOS-191127 entitled "The effect of bed roughness uncertainty on tidal stream power estimates for the Pentland Firth" has been accepted for publication in Royal Society Open Science subject to minor revision in accordance with the referee suggestions. Please find the referees' comments at the end of this email.

The reviewers and handling editors have recommended publication, but also suggest some minor revisions to your manuscript. Therefore, I invite you to respond to the comments and revise your manuscript.

- Ethics statement

- Data accessibility

It is a condition of publication that all supporting data are made available either as supplementary information or preferably in a suitable permanent repository. The data accessibility section should state where the article's supporting data can be accessed. This section

should also include details, where possible of where to access other relevant research materials such as statistical tools, protocols, software etc can be accessed. If the data has been deposited in an external repository this section should list the database, accession number and link to the DOI for all data from the article that has been made publicly available. Data sets that have been deposited in an external repository and have a DOI should also be appropriately cited in the manuscript and included in the reference list.

If you wish to submit your supporting data or code to Dryad (<http://datadryad.org/>), or modify your current submission to dryad, please use the following link:
<http://datadryad.org/submit?journalID=RSOS&manu=RSOS-191127>

- **Competing interests**

- **Authors' contributions**

- **Acknowledgements**

- **Funding statement**

Because the schedule for publication is very tight, it is a condition of publication that you submit the revised version of your manuscript before 26-Oct-2019. Please note that the revision deadline will expire at 00.00am on this date. If you do not think you will be able to meet this date please let me know immediately.

To revise your manuscript, log into <https://mc.manuscriptcentral.com/rsos> and enter your Author Centre, where you will find your manuscript title listed under "Manuscripts with Decisions". Under "Actions," click on "Create a Revision." You will be unable to make your

revisions on the originally submitted version of the manuscript. Instead, revise your manuscript and upload a new version through your Author Centre.

Once again, thank you for submitting your manuscript to Royal Society Open Science and I look

forward to receiving your revision. If you have any questions at all, please do not hesitate to get in touch.

Kind regards,
Anita Kristiansen
Editorial Coordinator
Royal Society Open Science
openscience@royalsociety.org

on behalf of Professor Ian Guymer (Associate Editor) and R. Kerry Rowe (Subject Editor)
openscience@royalsociety.org

Reviewer comments to Author:

Reviewer: 1

Comments to the Author(s)

Manuscript ID RSOS-191127 entitled "The effect of bed roughness uncertainty on tidal stream power estimates for the Pentland Firth" for Royal Society Open Science.

The manuscript describes a study examining the impact of varying bed roughness in a real-world tidal energy setting. It uses 2D shallow water model solutions to examine the impact on energy extraction through variation in bottom friction.

It is a useful result if very incremental, so in that sense – fits the purpose of the journal.

It is clearly written.

A number of points really should be followed up upon to consolidate the worth of the manuscript.

The results appear to suggest minimal impact of varying C_d – but the same value of C_d can have a difference effect in different model configurations (scale etc). How does this sensitivity compare?

I really struggle with 2D perspectives on anything other than the very broad scale. The energy extraction will be highly heterogeneous in the vertical. How important is this likely to be?

By the same token, does the C_d variation fall within the impact of moving to 3D simulation?

Further, what's the meaning of 3% variation in power production in the context of the model. Is it thought that the present model is less than 3% from "truth"?

Is there some evidence that the present result is extendable to other systems and why?

How does the conclusion compare with the varying predictions mentioned in the abstract?

Reviewer: 2

Comments to the Author(s)

This is a well written paper about tidal power in the Pentland Firth following a previous paper using idealised models. This paper uses more realistic depth-averaged shallow water tidal model

for the region. I do have some comments. The paper refers to power potential but it is not clear from the start that is about 'removable power' rather than 'extractable power' clarified later. Initially I thought it was about power extracted by turbines. I realise this cannot be simply undertaken but some estimate could be given, e.g. 50% based on some idealised cases. If it was 10% for example the exercise would be rather pointless. In the Conclusions powers of GW level are discussed but this is for removable power.

Going through the paper, in the Introduction it is stated that 'there is uncertainty associated with the correct value of bed roughness coefficient that should be applied.' It should be clear that this is associated with depth-averaged models used here. 3-D models with hydrostatic pressure are more accurate and indeed show that 2-D depth-averaged models with strong curvature, recirculating flows in the extreme, can be rather misleading (Stansby et al 2016). Soon after values of C_d for a rough surface are referred to but I think these are for steady flow. In oscillatory they can be different. This should be clarified.

In section 2a it is said that 'little change to the natural currents at the boundary occurs'. Little should be quantified in some way, e.g. 1% or 10% in change in tidal level or volume flux. 'little' means different things to different people!

I would like to see fig 1c enlarged so the fences can be more easily seen. Why not include in fig 1b?

In eq 2.4 is \bar{u} with or without the fence present?

The actual analysis of uncertainty is quite comprehensive and dense.

With these relatively minor points addressed I think the paper will be suitable for publication.

Stansby, P. Chini, N. and Lloyd, P. 2016 Oscillatory flows around a headland by 3D modelling with hydrostatic pressure and implicit bed shear stress comparing with experiment and depth-averaged modelling, Coastal Engineering 116, 1-14

Reviewer: 3

Comments to the Author(s)

Tidal stream power estimates often rely on the results of hydrodynamic numerical models that are calibrated using bed friction coefficients. This way of doing is subjected to high uncertainties. Interestingly, this paper evaluates the effect of these uncertainties on the resource assessment by considering different scenarios of turbine deployments.

The paper, written following the IMRAD structure, is well written and contains high quality figures. The methodology is clearly described and the conclusions are well supported by the results. I therefore strongly recommend its publication.

My only (minor) comment concerns the omission of modelling studies with non-uniform roughness coefficients (in the introduction, you suggest that, in existing studies, C_d is always uniform). As using a uniform C_d is not always applicable, especially in tidal sites where the seabed is highly heterogeneous, I wonder how is the seabed in the Pentland Firth. Showing a sedimentary map could help justifying the assumption $C_d = \text{constant}$.

Author's Response to Decision Letter for (RSOS-191127.R0)

See Appendix A.

Decision letter (RSOS-191127.R1)

26-Nov-2019

Dear Dr Kreitmair,

It is a pleasure to accept your manuscript entitled "The effect of bed roughness uncertainty on tidal stream power estimates for the Pentland Firth" in its current form for publication in Royal Society Open Science. The comments of the reviewer(s) who reviewed your manuscript are included at the foot of this letter.

Kind regards,
Lianne Parkhouse
Editorial Coordinator
Royal Society Open Science
openscience@royalsociety.org

on behalf of Professor Ian Guymer (Associate Editor) and Professor R. Kerry Rowe (Subject Editor)
openscience@royalsociety.org

Editor Comments to Author:

My only very minor query, is regarding the location of Figure 7. Whilst I believe that it is first

referenced in the "A Power Surfaces" section, it doesn't look right to see it placed mid-way through the References. Please consider amending this within your finalised paper during the proofing process.

Appendix A

Anita Kristiansen

Editorial Coordinator

RSOS-191127

Title: The effect of bed roughness uncertainty on tidal stream power estimates for the Pentland Firth

Authors: M.J. Kreitmair, T.A.A. Adcock, A.G.L. Borthwick, S. Draper, and T.S. van den Bremer

Dear Anita Kristiansen,

Thank you very much for your kind email concerning our manuscript titled “The effect of bed roughness uncertainty on tidal stream power estimates for the Pentland Firth” (RSOS-191127) which was submitted to *Royal Society Open Science* for possible publication. We have modified the paper taking full consideration of all the comments and suggestions from the Reviewers. Corresponding revisions are indicated by red font in the new version of our manuscript. We also provide a detailed list of the revisions, along with itemized responses to each comment and suggestion, which we believe have led to a significant improvement in the quality of the manuscript. Two copies of the pdf are submitted: one where changes to the manuscript are indicated in red, and the other the final version. We confirm that format, style and referencing style of the finalized manuscript should fit the requirements of the Journal.

We would like to thank you again for your detailed suggestions. Sincere thanks are also due to the anonymous reviewers for their very helpful comments.

With best regards.

Sincerely yours,

Dr Monika Kreitmair

Authors' Response

Reviewer #1 Comments

The manuscript describes a study examining the impact of varying bed roughness in a real-world tidal energy setting. It uses 2D shallow water model solutions to examine the impact on energy extraction through variation in bottom friction. It is a useful result if very incremental, so in that sense – fits the purpose of the journal. It is clearly written.

Response: We thank the reviewer for his/her insightful comments regarding the manuscript. We have attempted to address each of the comments, as per the detailed responses below.

A number of points really should be followed up upon to consolidate the worth of the manuscript. The results appear to suggest minimal impact of varying C_d – but the same value of C_d can have a difference effect in different model configurations (scale etc). How does this sensitivity compare?

Response: In this paper, we have solely considered parameter uncertainty and what effect it has within a particular model configuration (i.e. model scale, mesh refinement, etc.). Changes to the model would affect both the value of the calibrated bed roughness coefficient and the magnitude of the associated parameter uncertainty. For example, a finer mesh could resolve physical features which are treated by the roughness coefficient in a coarser mesh. There exists an interaction between the model accuracy and the parameter uncertainty. This subtle point has potentially interesting consequences. We have addressed this, as well as the issues raised in the next comment, in a few sentences in the conclusions dealing with limitations.

Neglecting the interplay between model and parameter uncertainty, the impact of changing the model configuration on the value of bed roughness coefficient used is likely to be comparable to the standard deviations in C_d used in the paper.

I really struggle with 2D perspectives on anything other than the very broad scale. The energy extraction will be highly heterogeneous in the vertical. How important is this likely to be? By the same token, does the C_d variation fall within the impact of moving to 3D simulation?

Response: Three-dimensionality is important of course, particularly at the location of the turbines or other obstacles where the wake and recirculation zones can cause significant increase in the value of the bed roughness coefficient (perhaps by an order of magnitude

over the background level, see e.g. Stansby, Chini, and Lloyd, 2016). The consequent vertical mixing due to secondary flows cannot be captured by depth-averaged models (Stansby, 2006). However, at the scale of the present study, vertical heterogeneity has an almost negligible effect, and so 2D models are sufficiently capable of simulating bulk flow through the head-driven channels of the Pentland Firth and providing a sensible estimate of the associated energy removal (Adcock, Draper, and Nishino, 2015). As the scale of the modelled domain decreases, the uncertainty in results stems increasingly from model discrepancy (i.e. the consequence of using an over-simplified model that cannot account for increasingly important three-dimensional features) rather than from parameter uncertainty (uncertainty associated with the correct value of a calibration parameter). We have added text to this effect in the Introduction and Conclusions of the revised manuscript.

Stansby P. K., Chini N., Lloyd P. (2016) Oscillatory flows around a headland by 3D modelling with hydrostatic pressure and implicit bed shear stress comparing with experiment and depth-averaged modelling. *Coastal Engineering*, 116:1-14.

Stansby P. K. (2006) Limitations of depth-averaged modelling for shallow wakes. *J. Hydraul. Eng.* 132:737–740

Adcock T. A. A., Draper S., Nishino T. (2015) Tidal power generation - A review of hydrodynamic modelling. *Proc IMechE Part A: J Power and Energy* 0(0):1-17

Further, what's the meaning of 3% variation in power production in the context of the model. Is it thought that the present model is less than 3% from "truth"?

Response: No, the variation referred to in the paper is the change in expected power as we move from considering a deterministic bed roughness coefficient to one with a probability distribution (i.e. comparing the mean of the power values for a distribution of Cd values to the power value calculated at the mean Cd) and is a measure as to how much uncertainty affects the model. This is stated in the first paragraph of section 3a.

Is there some evidence that the present result is extendable to other systems and why?

Response: Yes, there is analogous work in hydrological engineering science using a method called the derived distribution approach (Ang and Tang, 1975) that uses an analytic method to derive the probability distribution of a dependent variable from uncertainty in the independent variables. Example applications include flood frequency analysis by Eagleson (1972), Hebson and Wood (1982), Díaz-Granados et al. (1984), urban stormwater runoff by Chan and Bras (1979), subgrid hydrological processes in

atmospheric general circulation models (Entekhabi and Eagleson, 1989), rainwater interception (Ramírez and Senarath, 2000), and optimal water allocation (Perona et al. 2013). We have added text to the introduction mentioning the use of this approach in other fields.

Ang A. H.-S. and Tang W. H. (1975) *Probability concepts in engineering planning and design: Vol. 1, Basic principles*. Wiley, New York.

Chan S.-O. and Bras R. (1979) Urban stormwater management: Distribution of flood volumes. *Water Resources Research*, 15(2): 371-382.

Díaz-Granados M. A., Valdés J. B. and Bras, R. L. (1984) A physically-based flood frequency distribution. *Water Resources Research*, 20(7): 995-1002.

Eagleson P. S. (1972) Dynamics of flood frequency. *Water Resources Research*, 8(4): 878-898.

Entekhabi, D. and Eagleson P.S. (1989): Land surface hydrology parameterization for atmospheric General Circulation models including subgrid scale spatial variability. *Journal of Climate*, 2(8): 816-831.

Hebson C. and Wood E. F. (1982) A derived flood frequency distribution using Horton Order Ratios. *Water Resources Research*, 18(5): 1509-1518.

Perona P., Dürrenmatt D.J. and Characklis G.W. (2013) Obtaining natural-like flow releases in diverted river reaches from simple riparian benefit economic models. *Journal of Environmental Management*,

Ramírez J. A. and Senarath S. (2000) A statistical–dynamical parameterization of interception and land surface – atmosphere interactions. *Journal of Climate*, 13(22): 4050-4063.

How does the conclusion compare with the varying predictions mentioned in the abstract?

Response: This is a good point and we have added the following sentences to the Conclusions:

“Comparing this interval with the range of power estimates for the Pentland Firth from the Introduction, *i.e.* 0.62-9 GW, only a fraction of this range may be attributed to bed roughness uncertainty. The greater proportion is most likely due to differences in models used, which indicates the requirement for a coherent methodology in the tidal stream energy industry to allow for comprehensive assessments to be made.”

Reviewer #2 Comments

This is a well written paper about tidal power in the Pentland Firth following a previous paper using idealised models. This paper uses more realistic depth-averaged shallow water tidal model for the region. I do have some comments. The paper refers to power potential but it is not clear from the start that is about ‘removable power’ rather than ‘extractable power’ clarified later. Initially I thought it was about power extracted by turbines. I realise this cannot be simply undertaken but some estimate could be given, e.g. 50% based on some idealised cases. If it was 10% for example the exercise would be rather pointless. In the Conclusions powers of GW level are discussed but this is for removable power.

Response: We thank the reviewer for his/her comments. We have now added the word “removable” in the abstract to make clear the fact that we are considering removable power in this approach. Likewise, in the conclusions, the text has been changed to make clear that the results are for removable and not available or extractable power.

We are not sure what estimate is meant by the reviewer in the second half of the comment but have answered this assuming the he/she is asking about an estimate for the effect of bed roughness uncertainty on extractable rather than removable power. Assuming a linear relationship between the extractable and removable power (i.e. the amount of power that may be extracted is a constant fraction of the power removed from the channel), the proportional effect of bed roughness uncertainty would simply transfer to removable power. More complicated, non-linear relationships would require further analysis. Additional text to this effect has been added to the Conclusions in the revised manuscript.

Going through the paper, in the Introduction it is stated that ‘there is uncertainty associated with the correct value of bed roughness coefficient that should be applied.’ It should be clear that this is associated with depth-averaged models used here. 3-D models with hydrostatic pressure are more accurate and indeed show that 2-D depth-averaged models with strong curvature, recirculating flows in the extreme, can be rather misleading (Stansby et 2016). Soon after values of C_d for a rough surface are referred to but I think these are for steady flow. In oscillatory they can be different. This should be clarified.

Response: We have clarified that the bed roughness coefficient explored in the paper is that associated with depth-averaged models by adding text to the introduction to make this explicit. Regarding steady vs. oscillatory flow, the studies referenced in the paper modelled the Pentland Firth with oscillating head differences driving the flow through the strait.

In section 2a it is said that ‘little change to the natural currents at the boundary occurs’. Little should be quantified in some way, e.g. 1% or 10% in change in tidal level or volume flux. ‘little’ means different things to different people!

Response: We have added a sentence quantifying the effect.

I would like to see fig 1c enlarged so the fences can be more easily seen. Why not include in fig 1b?

Response: We have changed the figures to include the fences in 1b and hope this has made the images clearer.

In eq 2.4 is \bar{u} with or without the fence present?

Response: We assume the reviewer means bold u ? This is the depth-averaged velocity vector at the locations of enhanced roughness with fences present, i.e. increased roughness. The text immediately following equation (2.4) has been amended to make this clear.

The actual analysis of uncertainty is quite comprehensive and dense.

Response: Thank you.

With these relatively minor points addressed I think the paper will be suitable for publication.

Response: We thank the reviewer for the helpful comments. We hope the changes made have fully resolved all of the points made.

Reference

Stansby, P. Chini, N. and Lloyd, P. 2016 Oscillatory flows around a headland by 3D modelling with hydrostatic pressure and implicit bed shear stress comparing with experiment and depth-averaged modelling, Coastal Engineering 116, 1–14

Reviewer #3 Comments

Tidal stream power estimates often rely on the results of hydrodynamic numerical models that are calibrated using bed friction coefficients. This way of doing is subjected to high uncertainties. Interestingly, this paper evaluates the effect of these uncertainties on the resource assessment by considering different scenarios of turbine deployments.

The paper, written following the IMRAD structure, is well written and contains high quality figures. The methodology is clearly described and the conclusions are well supported by the results. I therefore strongly recommend its publication.

Response: Thank you. We greatly appreciate the positive response.

My only (minor) comment concerns the omission of modelling studies with non-uniform roughness coefficients (in the introduction, you suggest that, in existing studies, C_d is always uniform). As using a uniform C_d is not always applicable, especially in tidal sites where the seabed is highly heterogeneous, I wonder how is the seabed in the Pentland Firth. Showing a sedimentary map could help justifying the assumption $C_d = \text{constant}$.

Response: We agree that use of a uniform C_d is a problem with existing studies and that such an assumption may not be justified by variation in bed composition. In the paper, we identify a constant bed roughness coefficient as a potential source of uncertainty (technically model uncertainty). Due to the prevalent use of a single value of coefficient in tidal energy studies, in particular the Pentland Firth, we chose to explore the uncertainty resulting within a uniform C_d model and the true bed texture variability of the Pentland Firth is, to some degree, irrelevant. Text has been added in the Introduction and Conclusions to clarify the issue raised by applying a uniform C_d , and a sentence has been added after equation (2.3) to explain our use of a uniform C_d .